# Impact of Patient, Surgical, and Perioperative Factors on Discharge Disposition after Radical Cystectomy

**DOI:** 10.3390/cancers14215288

**Published:** 2022-10-27

**Authors:** Giovanni E. Cacciamani, Ryan S. Lee, Daniel I. Sanford, Wesley Yip, Jie Cai, Gus Miranda, Siamak Daneshmand, Monish Aron, Hooman Djaladat, Inderbir S. Gill, Mihir Desai

**Affiliations:** Catherine & Joseph Aresty Department of Urology, Keck Medicine of USC, University of Southern California, Los Angeles, CA 90007, USA

**Keywords:** radical cystectomy, discharge disposition, marital status, disparities, insurance

## Abstract

**Simple Summary:**

Radical cystectomy is a surgical procedure used in the treatment of bladder cancer. It can potentially cure a patient of their disease, but it is associated with risks including long hospital stays and high complication rates. Here, we evaluated how certain factors related to a patient and/or their surgery may impact their discharge to home versus to a rehabilitation facility after surgery. This study found that patients that are older, living alone prior to surgery, and/or have a major post-operative complication are more likely to be discharged to a rehabilitation facility. Recognizing these factors prior to surgery may help identify which patients may require discharge to a rehabilitation facility and can help patients and urologists better plan for the postoperative course.

**Abstract:**

Radical cystectomy (RC) is a complex procedure associated with lengthy hospital stays and high complication and readmission rates. We evaluated the impact of patient, surgical, and perioperative factors on discharge disposition following RC at a tertiary referral center. From 2012 to 2019, all bladder cancer patients undergoing RC at our institution were identified (*n* = 1153). Patients were classified based on discharge disposition: to home or to continued facility-based rehabilitation centers (CFRs, *n* = 180 (15.61%) patients). On multivariate analysis of patient factors only, age [Risk Ratio (RR): 1.07, *p* < 0.001)], single marital status (RR: 1.09, *p* < 0.001), and living alone prior to surgery (RR: 2.55, *p* = 0.004) were significant predictors of discharge to CFRs. Multivariate analysis of patient, surgical, and perioperative factors indicated age (RR: 1.09, *p* < 0.001), single marital status (RR: 3.9, *p* < 0.001), living alone prior to surgery (RR: 2.42, *p* = 0.01), and major post-operative (Clavien > 3) complications (RR: 3.44, *p* < 0.001) were significant independent predictors of discharge to CFRs. Of note, ERAS did not significantly impact discharge disposition. Specific patient and perioperative factors significantly impact discharge disposition. Patients who are older, living alone prior to surgery, and/or have a major post-operative complication are more likely to be discharged to CFRs after RC.

## 1. Introduction

Patient discharge disposition after surgery is a crucial aspect of perioperative care that impacts health care costs and it is listed as one of the measures of quality of care in the Center for Medicare and Medicaid Services (CMS) [1,2]. Identifying factors that predict discharge to continued facility-based rehabilitation centers (CFRs), such as skilled nursing facilities or acute rehabilitation units, may help improve perioperative protocols to facilitate surgeons’ and patients’ expectations [3] and guide proper resource allocation [4,5].

Radical cystectomy (RC) is the gold-standard surgical treatment for muscle-invasive bladder cancer [6]. RC is a complex procedure that frequently involves significant postoperative complications resulting in prolonged hospital stays and readmissions [7], placing a significant demand on the health care system [8,9]. Currently, there is limited data available on discharge disposition [10] and factors predicting discharge to CFRs after RC [11,12].

Herein, we aim to evaluate the impact of patient, surgical, and perioperative factors on discharge disposition

## 2. Materials and Methods

### 2.1. Population

Between September 2012 to March 2019, all bladder cancer patients undergoing radical cystectomy and urinary diversion with intent to cure were identified from our prospectively collected, IRB-approved institutional radical cystectomy database (HS-01B014). All procedures were performed by eight experienced surgeons. Open and robotic RC, continent (orthotopic and non-orthotopic), and incontinent urinary diversions were performed using our previously described techniques [13,14,15,16,17,18,19]. Baseline demographic, perioperative, and pathologic data were analyzed. Thirty- and ninety-day postoperative complications were classified according to the modified Clavien–Dindo classification [20]. Readmission was defined as any subsequent inpatient admission or unplanned visit occurring within 30 or 90 days from the day of discharge [7]. Post-operatively, the patients were managed per our institutional Enhanced Recovery After Surgery (ERAS) protocol as described previously [21] with some surgeon-specific differences [22]. Patients were recommended discharge to either home or to a CFR based on detailed physical and occupational therapy evaluation. 

### 2.2. Statistical Analysis

Patients were distributed into 2 groups based on discharge disposition after RC: to home with home health services) or to CFRs, which include skilled nursing facilities, acute rehabilitation units, and transitional care units.

A univariate analysis was performed to compare baseline differences in demographics, surgical factors, and perioperative outcomes between the two groups. Continuous and categorical variables were presented as mean and standard deviation (SD), and median and interquartile range (IQR), respectively. In the univariate analysis, Kruskal-Wallis, chi-squared (*X*^2^), and Fisher exact tests were used to compare continuous and categorical variables as appropriate. We then performed separate multivariable logistic regression models. The travel distance was determined by measuring the straight-line distance (in miles) between the patient’s residence ZIP code/postal code and the treatment institution’s ZIP code/postal code (ZIPDISTANCE FUNCTION; SAS software, version 9.2; SAS Institute Inc., Cary, NC, USA) 

With our predictive model analysis, we sought to identify independent factors associated with our main outcome of interest: discharge to CFRs. The multivariable model included variables previously found to be predictors of discharge to CFR and significant variables from our preliminary univariate analysis. Patient characteristics included in the multivariable model were age, gender, body mass index (BMI), race, ASA score, Charlson Comorbidity Index (CCI), smoking habits, marital status, living situation before surgery, type of primary health insurance, and neoadjuvant chemotherapy (NACh). Surgical and perioperative variables included in the multivariable model were ERAS protocol, surgical approach (open or robot-assisted), type of urinary diversion, and occurrence of in-hospital complications.

A two-tailed test with *p* < 0.05 was considered statistically significant. All statistical analyses were performed using SPSS v.24.0 (SPSS Inc., Chicago, IL, USA). Data and methods were reported according to the most recent guidelines for reporting statistics for clinical research in urology [23].

## 3. Results

### 3.1. Baseline Demographics

A total of 1153 patients were included in this study. Demographic data are detailed in Table 1. Median (IQR) age was 71.0 (63.0–77.0) years and BMI was 26.9 (23.7–30.3) kg/m^2^. A total of 928 (80.49%) patients were male, 938 (81.42%) patients had ASA score of III or IV, 361 (31.31%) patients received neoadjuvant chemotherapy, and 395 (34.26%) patients presented with pT ≥ 3 disease. In our cohort, 836 (72.51%) patients were married and 220 (19.25%) lived alone prior to surgery. Most patients (58.63%) in our cohort had government (i.e., Medicare, Tricare) insurance. A detailed description of insurance types is demonstrated in Table 1.

### 3.2. Perioperative Outcomes

Perioperative outcomes are detailed in Table 2. 763 (66.18%) patients underwent open radical cystectomy compared to 390 (33.82%) patients who underwent robot-assisted radical cystectomy. The mean (SD) operative time was 6.0 (1.7) hours. Orthotopic neobladder and an incontinent urinary diversion were performed in 587 (50.91%) and 527 (45.71%) patients, respectively. Median (IQR) intraoperative estimated blood loss (EBL) was 400 (200–400) mL, and 220 (19.08%) patients required perioperative blood transfusions. 381 (33.10%) and 98 (8.51%) patients experienced minor (Clavien I-II) and major (Clavien III-IV) complications during their hospital course, respectively. Mean (SD) length of stay was 6.3 (5.0) days, and 90-day readmission and mortality were seen in 354 (30.70%) and 53 (4.60%) patients, respectively.

### 3.3. Discharge Disposition

Figure 1 depicts the annual distribution of discharge dispositions. A total of 180 (15.61%) patients were discharged to CFRs. On univariate analysis of patient characteristics, age (68.0 vs. 75.8 years, *p* < 0.001), gender (13.69% of males vs 23.56% of females, *p* < 0.001), ASA (*p* =0.005), CCI (*p* = 0.002), preoperative albumin level (26.3 vs. 25.1 g/dL, *p* = 0.05), baseline hemoglobin (12.1 vs. 11.6 g/dL, *p* = 0.04), baseline creatinine (1.2 vs 1.2 mg/dL, *p* = 0.05), marital status (*p* < 0.001), living alone before surgery (*p* < 0.001), living alone after surgery (*p* < 0.001), and insurance type (*p* = 0.01) were significantly different in patients discharged home compared to patients discharged to CFRs. On univariate analysis of perioperative variables, urinary diversion type (*p* < 0.001), perioperative (*p* = 0.023) and post-operative (*p* = 0.001) transfusions, complications during the hospital course (*p* < 0.001), and hospital length of stay (LOS) (5.8 vs. 7.0 days, *p* < 0.001) were significantly different. Mortality rate at 30 days (0.72% vs 2.78%, *p* = 0.027) and 90 days (2.88% vs. 13.89%, *p* < 0.001) as well as readmission rate at 30 days (17.99% vs 25.00%, *p* = 0.03) and 90 days (24.63% vs. 38.89%, *p* = 0.011) were also significantly different between the two groups. Of note, no significant difference was seen in the distance traveled for care between the two groups (*p* = 0.89).

Multivariate analysis of combined patient, surgical and perioperative factors (Figure 2) indicated that age (RR: 1.09, *p* < 0.001), single marital status (RR: 3.9, *p* < 0.001), living alone prior to surgery (RR: 2.42, *p* = 0.01) and major complications during the hospital course (RR: 3.44, *p* < 0.001) were significant predictors of discharge to CFRs. Surgical approach and type of urinary diversion were no longer significant risk factors when assessing patient and surgical factors on multivariate analysis.

## 4. Discussion

The last decade has brought several changes in the perioperative management of RC patients, especially with the incorporation of ERAS protocols, which have been shown to reduce hospital length of hospital stay, time to first flatus, and time to resumption of a regular diet [24,25,26]. ERAS protocols are also associated with improved quality-of-life measures during the recovery process and a decrease in overall costs [27]. Complication and readmission rates at 90 days after surgery, however, have not significantly changed with ERAS protocols [28]. While an abundance of research has evaluated perioperative management of radical cystectomy patients, there remains a paucity of modern data with conflicting results regarding patient discharge disposition [10,11,12,29].

Discharge to CFRs after a major abdominopelvic surgery is associated with a worse prognosis and overall survival [30]. Moreover, readmissions from these facilities are common, costly, and often avoidable. Approximately 23.5% of Medicare beneficiaries are readmitted to a hospital within 30 days of discharge to a skilled nursing facility, and 78% of these readmissions are potentially avoidable [31]. Taub et al. first reported the increasing proportion of discharge to subacute care facilities after cystectomy using data from the Nationwide Inpatient Sample. From 1988 to 2000, discharge to CFRs increased from 5.3 to 13.2% [10]. In a retrospective review of a single-institution series of 445 patients from 2004 to 2007, the main factors associated with discharge to a CFR after RC were older age, lower preoperative fitness level, and longer length of stay. Of note, in our cohort, patients discharged to a CFR had a higher mortality rate within 90-days of surgery (13.89% vs. 2.88%) [12]. Several factors have been identified as predictors of discharge to CFRs after cystectomy, including frailty, skeletal mass, and fat mass [11,29].

A better understanding of predictors of discharge to CFRs, specifically those that are modifiable, could help guide preoperative counseling and discharge planning. Our analysis of 1153 patients undergoing RC for bladder cancer in a tertiary referral center suggests that several patients and perioperative factors significantly impact discharge disposition. Importantly, we also found that patients discharged to CFRs had 90-day readmission and mortality rates similar to the rates reported by Aghazadeh et al. [12].

We found age to be a significant factor for discharge to a CFR. However, age is a non-modifiable factor that will likely continue to impact discharge disposition as most patients diagnosed with and undergoing RC for bladder cancer are older [32,33,34] with an average age of 73 years at diagnosis [35]. Care for the post-cystectomy patient can be complex and requires additional support at home [36]. In a qualitative study of bladder cancer patient experiences, patients valued support from friends and family, especially when learning new techniques such as catheterization [37]. Thus, it is not surprising that patients who are not married and/or living alone are more likely to be discharged to a CFR. Similarly, elderly patients living alone prior to both emergent and elective abdominal procedures are more likely to be discharged to subacute facilities [38,39]. In contrast to a study of 6460 RC patients in the Premier Perspectives Database by Nayak et al., comorbidities were not a predictor of discharge to CFRs in our analysis [40]. Thus, our association of age and discharge disposition may be more related to functional and performance status, such as frailty, rather than medical complexity. It has been reported that tobacco smoking impacts perioperative complications [41] and overall survival [42] after RC, but its impact on discharge disposition has not yet been evaluated. Our results did not find an association between smoking habits and discharge disposition after RC. The patient-specific results of our study can potentially guide preoperative counseling to manage expectations and to start discharge planning early (as part of our ERAS protocol this is conducted on POD1).

Of the surgery and perioperative factors that were significant in our analysis, patients undergoing non-orthotopic diversions were more likely to be discharged to a CFR. In our cohort, 89.44% of patients undergoing neobladder formation were discharged home. Although an open approach was associated with discharge to CFRs on univariate analysis, it is confounded by the fact that patients undergoing an open procedure were often older than those undergoing robotic surgery. The median length of stay in our cohort was 5 days, and patients with prolonged length of stay (>5 days) were more likely to be discharged to a CFR. This could be associated with non-medical factors causing prolonged hospitalization, such as locating a CFR. Lastly, patients who experienced major complications (Clavien III–IV) during their admission were also more frequently discharged to a facility, which is likely related to their overall complexity of care and recovery from complications. Our prior work has found that intraoperative transfusions were associated with 30-day complication rates in patients undergoing radical cystectomy with an ERAS protocol [43]. As such, reduction in major perioperative complications may be the only potential modifiable factor in discharge disposition and should remain a top focus of ERAS protocols.

Even though our study suggests that most factors that affect discharge disposition may not be modifiable, this information is useful in optimizing care even in patients who are likely to be discharged to CFR’s. ERAS protocols [44,45,46] should include a pre-operative checklist that determines the likelihood of discharge to a CFR pre-operatively so that the appropriate facility can be assigned to minimize unnecessarily prolonged hospital stays. CFR’s should be educated with the specific needs and early complication identification of radical cystectomy patients to reduce the probability of readmission from these facilities.

Our study has several limitations. Although our bladder cancer database is prospectively maintained, this study was a retrospective review and analysis. The analysis included patients who underwent RC through different approaches (open and robotic), and only one surgeon at our institution routinely performs both. Moreover, as demonstrated in a previously published internal audit from our institution, patients undergoing open RC were more compliant with ERAS protocol than those undergoing robotic RC [22]. Lastly, several confounding factors that were significant predictors of discharge disposition in other studies, such as frailty and performance status, were not available for inclusion in our analysis.

## 5. Conclusions

Patient and perioperative factors impact discharge disposition after radical cystectomy for bladder cancer. Our results suggest that predictors of discharge to CFRs seem to be nonmodifiable and can be identified prior to surgery (i.e., age, marital status, living situation) except for the presence of major complications during the post-operative hospital course. Thus, recognizing these nonmodifiable patient factors preoperatively can help identify which patients may require discharge to a CFR. By providing proper patient education, both patients and urologists can better plan for the postoperative course and anticipate discharge expectations and needs appropriately.

## Figures and Tables

**Figure 1 cancers-14-05288-f001:**
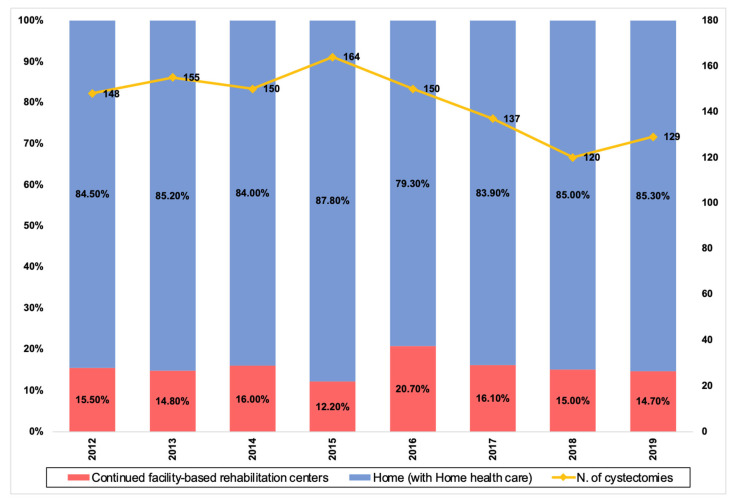
Trend over time of RC performed in our institution (2012–2019) after the implementation of our ERAS protocol and percentage of patients discharged home and to CFRs.

**Figure 2 cancers-14-05288-f002:**
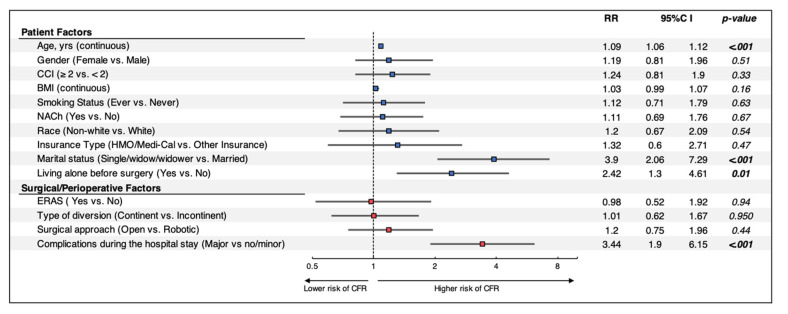
Multivariable regression models assessing the impact of patient, surgical, and perioperative factors on discharge disposition after RC. Patients without ERAS information were excluded from further analysis. *RR: Risk Ratio; CI: Confidence Interval; CCI: Charlson Comorbidity Index; BMI: body mass index; LOS: length of stay; NACh: Neoadjuvant Chemotherapy; ERAS: Enhanced Recovery After Surgery; CFR: Continued Facilities-based Rehabilitation Center.*

**Table 1 cancers-14-05288-t001:** Perioperative characteristics.

	Total	Home (with Home Care)	Continued Facility-Based Rehabilitation Centers	*p*-Value
**Patients, *n* (%)**	1153 (100.00%)	973 (84.39%)	180 (15.61%)	
**Age, yrs., mean (SD)/median (IQR)**	70.0 (10.5)/71.0 (63.0–77.0)	68.9 (10.4)/70.0 (62.0–76.0)	75.8 (9.0)/76.0 (71.0–82.0)	<0.001
**BMI, Kg/m², mean (SD)/median (IQR)**	27.4 (5.2)/26.9 (23.7–30.3)	27.4 (5.0)/26.9 (23.9–30.3)	27.0(5.9)/26.6 (22.1–30.8)	0.17
**Gender**				
**Male, *n* (%)**	928 (80.49%)	801 (82.32%)	127 (70.56%)	<0.001
**Female, *n* (%)**	225 (19.51%)	172 (17.68%)	53 (29.44%)	
**ASA score, mean (SD))/median (IQR)**				
**ASA 1–2, *n* (%)**	214 (18.58%)	194 (19.94%)	20 (11.74%)	0.005
**ASA 3–4, *n* (%)**	938 (81.42%)	779 (80.06%)	159 (88.83%)	
**CCI, mean (SD)/Median (IQR)**	1.4 (1.5)/1.0 (0.0–2.0)	1.3 (1.5)/1.0 (0.0–1.0)	1.7 (1.6)/1.0 (0.0–3.0)	0.001
**CCI 0, *n* (%)**	434 (37.6%)	385 (39.61%)	49 (27.22%)	0.005
**CCI 1, *n* (%)**	260 (22.57%)	216 (22.22%)	44 (24.44%)	
**CCI >/= 2, *n* (%)**	458 (39.76%)	371 (38.17%)	87 (48.33%)	
**Smoking Status**				
**Ever**	811 (70.34%)	686 (70.50%)	125 (69.44%)	0.79
**Never**	342 (29.66%)	237 (29/50%)	55 (30.56%)	
**Preoperative Albumin mean (SD)/Median (IQR)**	26.1 (20.0)/38.0 (0.0–42.0)	26.3 (20.1)/39.0 (0.0–42.0)	25.1 (19.5)/37.0 (0.0–41.0)	0.05
**Diabetes, *n* (%)**	253 (21.96%)	207 (21.30%)	46 (25.56%)	0.2
**Pulmonary comorbidity *n* (%)**	185 (16.06%)	146 (15.02%)	39 (21.67%)	0.035
**Cardiac comorbidity *n* (%)**	24 (2.08%)	18 (1.85%)	6 (3.33%)	0.25
**Renal comorbidity *n* (%)**	132 (11.46%)	106 (10.91%)	26 (14.44%)	0.2
**Baseline Hgb, mean (SD)/Median (IQR)**	12.0 (2.9)/12.0(10.2–13.6)	12.1 (3.0)/12.0 (10.3–13.7)	11.6 (2.0)/11.8 (9.9–13.1)	0.04
**Baseline Creatinine, mean, (SD)/Median (IQR)**	1.2 (0.7)/1.1 (0.9–1.4)	1.2 (0.7)/1.1 (0.9–1.4)	1.2 (0.6)/1.0 (0.9–1.4)	0.05
**NACH, *n* (%)**	361 (31.31%)	316 (32.48%)	45 (24.00%)	0.054
**Marital Status**				
**Married, *n* (%)**	836 (72.51%)	766 (78.73%)	70 (38.89%)	<0.001
**Single/Widowed, *n* (%)**	317 (27.49%)	207 (21.27%)	110 (61.1%)	
**Living status Before Surgery**				
**Alone, *n* (%)**	220 (19.25%)	133 (13.78%)	87 (48.88%)	<0.001
**Not Alone, *n* (%)**	923 (80.75%)	832 (86.22%)	91 (51.12%)	
**Living status After Surgery**				
**Alone, *n* (%)**	234 (20.58%)	81 (8.46%)	153 (85.47%)	<0.001
**Not Alone, *n* (%)**	903 (79.42%)	877 (91.54%)	26 (14.53%)	
**Race**				
**White American, *n* (%)**	955 (82.83%)	800 (82.22%)	155 (86.11%)	0.24
**Non-White American, *n* (%)**	198 (17.7%)	173 (17.78%)	25 (13.89%)	
**Insurance type**				
**Private (includes PPO, LOA, commercial), *n* (%)**	204 (17.69%)	171 (17.57%)	33 (18.33%)	0.01
**HMO, *n* (%)**	217 (18.82%)	201 (20.66%)	16 (8.89%)	
**Government (includes Medicare, Tricare)**	676 (58.63%)	553 (56.83%)	123 (68.33%)	
**Medi-Cal, *n* (%)**	19 (1.65%)	17 (1.75%)	2 (1.11%)	
**Cash, *n* (%)**	19 (1.65%)	16 (1.64%)	3 (1.67%)	
**Miscellaneous, *n* (%)**	18 (1.56%)	15 (1.54%)	3 (1.67%)	
**Pathologic Tumor Stage**				
**pT ≤ 2**				0.14
**pT0, *n* (%)**	190 (16.48%)	169 (17.37%)	21 (11.67%)	
**pTIS, *n* (%)**	171 (14.83%)	146 (15.01%)	25 (13.89%)	
**pTa, *n* (%)**	39 (3.38%)	34 (3.49%)	5 (2.78%)	
**pT1, *n* (%)**	148 (12.84%)	130 (13.36%)	18 (10.00%)	
**pT2a, *n* (%)**	103 (8.93%)	89 (9.15%)	14 (7.78%)	
**pT2b, *n* (%)**	107 (9.28%)	89 (9.15%)	18 (10.00%)	
**pT ≥ 3**				
**pT3a, *n* (%)**	170 (14.74%)	138 (14.18%)	32 (17.78%)	
**pT3b, *n* (%)**	105 (9.11%)	79 (8.12%)	26 (14.44%)	
**pT4a, *n* (%)**	116 (10.06%)	96 (9.87%)	20 (11.11%)	
**pT4b, *n* (%)**	4 (0.35%)	3 (0.31%)	1 (0.56%)	
**pNstage**				
**pN0, *n* (%)**	896 (77.71%)	761 (78.21%)	135 (75.00%)	0.06
**pN1, *n* (%)**	77 (6.68%)	65 (6.68%)	12 (6.67%)	
**pN2, *n* (%)**	158 (13.70%)	130 (13.36%)	28 (15.56%)	
**pN3, *n* (%)**	5 (0.43%)	5 (0.51%)	0	
**pNx, *n* (%)**	17 (1.47%)	12 (1.23%)	5 (2.78%)	
**pM stage**				
**pM0, *n* (%)**	1153 (100.0%)	973 (100%)	180 (100%)	1

**Table 2 cancers-14-05288-t002:** Surgical and Perioperative outcomes.

	Total	Home (with Home Care)	Continued Facility-Based Rehabilitation Centers	*p*-Value
**Patients, *n* (%)**	1153	973 (84.39%)	180 (15.61%)	
**Enhanced recovery after surgery setting**				
**ERAS, *n* (%)**	830 (89.63%)	705 (90.04%)	125 (87.41%)	0.37
**Non-ERAS, *n* (%)**	96 (10.37%)	78 (9.96%)	18 (12.59%)	
**Not reported, *n* (%)**	227	190	37	
**Surgical Approach**				
**Robotic, *n* (%)**	390 (33.82%)	328 (33.71%)	62 (34.44%)	0.864
**Open, *n* (%)**	763 (66.18%)	645 (66.29%)	118 (65.56%)	
**Type of Urinary Diversion**				
**Incontinent, *n* (%)**	527 (45.71%)	416 (42.75%)	111 (61.67%)	**<0.001**
**Continent–non-orthotopic, *n* (%)**	39 (3.38%)	32 (3.29%)	7 (3.89%)	
**Continent–orthotopic, *n* (%)**	587 (50.91%)	525 (53.96%)	62 (34.44%)	
**Operative Time, hours (SD)/median (IQR)**	6.0 (1.7)/5.9 (4.8–7.1)	6.1 (1.7)/4.9 (4.8–7.1)	6.0 (1.7)/5.7 (4.8–7.0)	0.36
**EBLs, ml (SD)/median (IQR)**	400.0 (200.0–600.0)	459.1 (354.4)/375.0 (200.0–600.0)	533.4 (832.6)/400 (250.0–600.0)	0.44
**Perioperative Transfusions, *n* (%)**	220 (19.08%)	174 (17.88%)	46 (25.56%)	**0.023**
**Postoperative transfusions, *n* (%)**	240 (20.82%)	186 (19.12%)	54 (30.00%)	**0.001**
**Perioperative complications**				
**In-hospital Complication (within hospital stay)**				
**No complications**	672 (58.38%)	594 (61.17%)	78 (43.33%)	**<0.001**
**Minor, *n* (%)**	381 (33.10%)	310 (31.93%)	71 (39.44%)	
**Major, *n* (%)**	98 (8.51%)	67 (6.90%)	31 (17.22%)	
**Transfusion *n* (%)**	380 (32.96%)	299 (30.73%)	81 (45.00%)	**<0.001**
**Cardiac system *n*, (%)**	128 (11.10%)	97 (9.97%)	31 (17.22%)	**0.007**
**Pulmonary system *n*, (%)**	39 (3.38%)	22 (2.26%)	17 (9.44%)	**<0.001**
**Gastrointestinal system *n*, (%)**	326 (28.27%)	269 (27.65%)	57 (31.67%)	0.28
**Uro-Genital system, *n* (%)**	269 (23.33%)	227 (23.33%)	42 (23.33%)	1
**Neuronal System *n*, (%)**	62 (5.38%)	53 (5.45%)	9 (5.00%)	1
**Infections, *n* (%)**	376 (32.61%)	299 (30.73%)	77 (42.78%)	**0.002**
**Overall, 30 days complications, *n* (%)**	705 (61.14%)	568 (58.38%)	137 (76.11%)	**<0.001**
**Minor, *n* (%)**	548 (47.53%))	451 (46.35%)	97 (53.89%)	**<0.001**
**Major, *n* (%)**	154 (13.36%)	114 (1.72%)	40 (22.22%)	
**Overall, 31–90 days complications, *n* (%)**	371 (32.18%)	310 (31.86%)	61 (33.89%)	0.6
**Minor, *n* (%)**	250 (21.68%)	211 (21.69%)	39 (21.67%)	0.31
**Major, *n* (%)**	101 (8.76%)	80 (8.22%)	21 (11.67%)	
**Overall, 90 days complications, *n* (%)**	843 (73.11%)	692 (71.12%)	151 (83.89%)	**<0.001**
**Minor, *n* (%)**	598 (51.86%)	501 (51.49%)	97 (53.89%)	**<0.001**
**Major, *n* (%)**	236 (20.47%)	182 (18.71%)	54 (30.00%)	
**LOS, days, mean (SD)/median (IQR)**	6.3(5.0)/5.9 (4.8–7.0)	5.8 (3.9)/5.0 (4.0–7.0)	7.0 (5.0–9.5)	**<0.001**
**LOS <= 5days *n*, (%)**	459 (39.81%)	430 (44.19%)	29 (16.11%)	**<0.001**
**LOS > 5days *n*, (%)**	694 (60.19%)	543 (55.81%)	151 (83.89%)	
**Mortality rate**				
**30 days mortality *n*, (%)**	12 (1.04%)	7 (0.72%)	5 (2.78%)	**0.027**
**90 days mortality *n*, (%)**	53 (4.60%)	28 (2.88%)	25 (13.89%)	**<0.001**
**Readmission rate**				
**30 days readmission, *n* (%)**	220 (19.08%)	17 (17.99%)	45 (25.00%)	**0.03**
**90 days readmission, *n* (%)**	354 (30.70%)	284 (24.63%)	70 (38.89%)	**0.011**

## Data Availability

The data presented in this study are available on request from the corresponding author. The data are not publicly available to maintain HIPAA Compliance.

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
