# Peer review of "Impact of Patient, Surgical, and Perioperative Factors on Discharge Disposition after Radical Cystectomy"

_cancers, 2022, doi:10.3390/cancers14215288_

Round 1
Reviewer 1 Report
The manuscript by Cacciamani studies factors on discharge disposition after radical cystectomy with the aim to identify modifiable factors to facilitate discharge planning using data from from a high-volume medical center. The analysis is retrospective in nature and highlighted some factors that predisposed patient for discharge to facilities. Findings from this study could potentially help better planning to decrease unnecessary hospital stay and therefore reduce the health care cost. Minor comment Figure 1 - It seems the authors wanted to present the precetage using bar graph and bars are missing from the figure. Please revise.Author Response
Dear Editor,
we have edited the figure n. 1 as requested by Rev 1.
Thanks
Daniel Sanford
Reviewer 2 Report
The cohort describe the incidence and variables associated with postoperative discharge after radical cystectomy to continued facility-based rehabilitation centers (CFRs) in a large referral cancer center during the years 2012-2019. 180 (15.6%) patients were discharge to CFRs after surgery with predictors by multivariate analysis includes – age, single marital status, living alone prior to surgery and major complications during the hospital course. Interesting status of ERAS use, type of surgery and type of diversion were not associated with discharge to CRFs by multivariate analysis, while LOS and diversion type were significant for CRFs discharge according to univariate analysis. The first 3 variable obviously can not be change while the last can.
The data regarding the predictors for CRFs discharge is limited in the literature, but the use of it is important for patients' education, institutes perioperative protocol and health care system resources.
To improve the paper, I suggest for the following:
1. What variables were associated with major complications rate following surgery?
2. Some papers describe a lower complications rate for the use of ERAS, please add it to the discussion.
Ahmadi H, Daneshmand S. Association between use of ERAS protocols and complications after radical cystectomy. World J Urol. 2022 Jun;40(6):1311-1316.
Xiao J, Wang M, He W, Wang J, Yang F, Ma XY, Zang Y, Yang CG, Yu G, Wang ZH, Ye ZQ. Does Postoperative Rehabilitation for Radical Cystectomy Call for Enhanced Recovery after Surgery? A Systematic Review and Meta-analysis.
.Curr Med Sci. 2019 Feb;39(1):99-110.
Vlad O, Catalin B, Mihai H, Adrian P, Manuela O, Gener I, Ioanel S. Enhanced recovery after surgery (ERAS) protocols in patients undergoing radical cystectomy with ileal urinary diversions: A randomized controlled trial. Medicine (Baltimore). 2020 Jul 2;99(27):e20902.
Author Response
Dear reviewer,
We thank you for your helpful comments on our manuscript. Please find our responses below:
- While not directly studied in the present study, we have previously assessed factors associated with complications following surgery. Intraoperative transfusions were associated with 30-day complication rates in patients undergoing radical cystectomy with an ERAS protocol (Zanfield et al. SAGE Journals 2018). We have included this reference in our discussion.
2. We thank the reviewer for pointing out these relevant and informative research studies. We have included the citations in our discussion.
Reviewer 3 Report
Dear Authors,
This paper adresses psotoperative care of RC patients which I think is a valuable addition to our current practice.
I don't have any furhter reccomendations
Author Response
We are grateful for the reviewer's time and comments on our manuscript. We greatly appreciate the feedback and look forward to implementing our findings into clinical practice.